# E2Usd: Efficient-yet-effective Unsupervised State Detection for Multivariate Time Series

## ABSTRACT

Cyber-physical system sensors emit multivariate time series (MTS) that monitor physical system processes. Such time series generally capture unknown numbers of states, each with a different duration, that correspond to specific conditions, e.g., "walking" or "running" in human-activity monitoring. Unsupervised identification of such states facilitates storage and processing in subsequent data analyses, as well as enhances result interpretability. Existing state-detection proposals face three challenges. First, they introduce substantial computational overhead, rendering them impractical in resource-constrained or streaming settings. Second, although state-of-the-art (SOTA) proposals employ contrastive learning for representation, insufficient attention to false negatives hampers model convergence and accuracy. Third, SOTA proposals predominantly only emphasize offline non-streaming deployment, we highlight an urgent need to optimize online streaming scenarios. We propose E2Usd that enables efficient-yet-accurate unsupervised MTS state detection. E2Usd exploits a Fast Fourier Transform-based Time Series Compressor (fftCompress) and a Decomposed Dual-view Embedding Module (ddEM) that together encode input MTSs at low computational overhead. Additionally, we propose a False Negative Cancellation Contrastive Learning method (fnccLearning) to counteract the effects of false negatives and to achieve more cluster-friendly embedding spaces. To reduce computational overhead further in streaming settings, we introduce Adaptive Threshold Detection (adaTD). Comprehensive experiments with six baselines and six datasets offer evidence that E2Usd is capable of SOTA accuracy at significantly reduced computational overhead. Our code is available at http://bit.ly/3rMFJVv.

## 1 INTRODUCTION

In Cyber-Physical Systems (CPSs) [23, 29, 41], sensors monitor physical processes continuously, generating streams of Multivariate Time Series (MTS) data. This raw data, often complex and devoid of immediate interpretability, requires human labors to discern underlying "states" that correspond to specific conditions. For instance, consider an MTS corresponding to a dance routine as depicted in Fig. 1. The MTS is collected using four accelerometers situated in the dancer's arms and legs, capturing transitions between states that can be labeled "walk", "run", "jump", "kick", and "left hop". The aim of *state detection* is to segment the MTS into a sequence of concise segments and assign each segment a state. Segments that share similar characteristics should be assigned the same state. In Fig. 1, the first and last segments exhibit similar fluctuations and are consequently assigned the same state, "walk".

Supervised state detection [12, 19, 27] requires known segments of an MTS and their labels, which are often not available. Thus, there has been a growing interest in *unsupervised state detection* (USD) [16, 20, 25, 26, 33, 36]. USD is capable of identifying distinct states in an MTS directly, without relying on known segments and

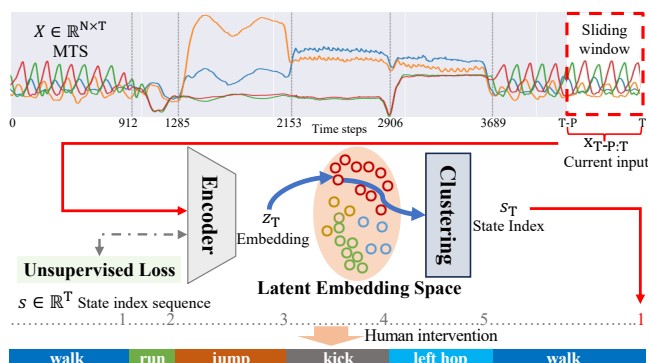

**Fig. 1: An example of unsupervised state detection on MTS.**

their labels. Once their USD process is completed, minimal human intervention is needed to assign a semantic label to each detected state. As depicted in Fig. 1, USD often uses clustering to associate each time step with a sufficiently similar already seen state or a new state if no similar state has been seen. Consecutive time steps with the same state are merged to form a segment. In the dance example, developers are not required to know the number and types of dance states beforehand. Instead, they can identify and label the dance states based on the USD results. This level of flexibility is particularly attractive in open-ended detection tasks, e.g., as found in cyber-attacks [9], web application behavior [21], and beyond.

A commonly used USD approach [20, 26, 36], as depicted in Fig. 1, involves two stages. An initial encoding stage projects input data, acquired using a sliding window, into a latent embedding; then, a clustering stage identifies the state of this latent embedding. By moving the sliding window as data arrives, it is possible to support real-time detection of states. While recent advances in deep learning (DL) have improved the initial MTS data encoding [26, 36], DL-based USD methods excel at capturing intricate MTS features, thus enhancing the subsequent clustering process. However, three main challenges remain.

**C1 (Resource-Intensive Architectures)**. The intricate architectures, particularly those of DL-based MTS encoders [26, 36], incur substantial computational and storage overheads. This precludes the deployment of such USD models on devices with limited resources, which occur frequently in practice.

**C2 (False Negative Sampling of Unsupervised Contrastive Learning)**. State-of-the-art (SOTA) learning proposals for MTS encoders are rooted in unsupervised contrastive learning [36] and aim to maximize the similarity between similar samples (from consecutive windows) and to minimize the similarity between dissimilar ones (from distant windows). However, this approach can be prone to false negative sampling due to its idealized assumption that distant windows have distinct states. Ensuring the robustness of unsupervised contrastive learning at forming a clustering-friendly embedding space is an important concern.

**C3 (Suboptimal Methodology on Streaming Scenarios).** While current studies [25, 26, 36] focus primarily on offline, non-streaming USD, there is a critical need for optimization for online deployments. Unconditional invocation of a USD model for all windowed MTS data can cause redundant clustering computations. Thus, more efficient strategies for streaming use are needed.

In this study, we present E2Usd, an efficient-yet-effective model addressing these challenges in unsupervised MTS state detection. To tackle **C1**, E2Usd includes a compact embedding method featuring two key strategies. First, it utilizes a Fast Fourier Transform-based Time Series Compressor (fftCompress) to selectively retain essential frequency components, discarding noisy ones. This reduces the computational overhead of subsequent operations. Second, to strike a balance between feature extraction capacity and model simplicity, E2Usd advocates a return to the original nature of a time series by decomposing compressed MTS into trend and seasonal components followed by a simple but sufficiently effective dual-view DL embedding module (named ddEM). This eliminates reliance on complex end-to-end DL architectures.

To tackle **C2**, we propose a False Negative Cancellation Contrastive Learning method (fnccLearning) tailored to mitigate false negative sampling in SOTA contrastive learning for USD. fnccLearning introduces a novel negative sampling scheme, where selecting genuinely false negatives is carried out by taking into account both the trend and seasonal similarities between paired samples. Moreover, instead of considering individual windowed samples, we take a holistic approach by harnessing groups of consecutive samples for similarity computation in the negative aspect, thereby ensuring consistent embedding within the same state. This unique treatment is reflected in the overall fnccLearning loss.

To tackle **C3**, we present Adaptive Threshold Detection (adaTD), which aims to reduce the number of clustering operations in online USD by first assessing the similarity between the currently windowed MTS data and the data in the preceding window and then deciding whether to perform clustering on the current windowed MTS data. A customized adaptive threshold, based on a simple and effective similarity metric is proposed to determine the similarity sufficiency. In experiments, E2Usd achieves the best accuracy while using only **4%** of the total and **1%** of the trainable parameters when compared to the SOTA method, while also achieving the lowest processing time among all competitors.

The primary contributions are as follows.

- We propose a compact MTS embedding method, comprising (i) fftCompress for retaining essential temporal information while mitigating noise for simplified time series representation and (ii) ddEM, which enables dual-view embedding of trend and seasonal components in MTS, effectively integrating traditional and modern methodologies (Section 3.1).
- We propose fnccLearning, aimed at mitigating the likelihood of false negatives in unsupervised contrastive learning for MTS state detection. This is achieved by a unique treatment of potential negative pairs exhibiting the lowest similarities (Section 3.2).
- We devise the adaTD scheme tailored for streaming USD. By comparing the current windowed MTS data to the preceding window, adaTD reduces clustering operations based on an adaptive similarity threshold (Section 3.3).

- We study E2Usd on six datasets while considering six baselines, providing evidence of SOTA accuracy and substantial computational costs reduction. We also provide evidence of practical applicability by deploying E2Usd on an STM32 MCU (Section 4).

Besides, Section 2 provides necessary background information, Section 5 reviews related work, and Section 6 concludes the paper.

## 2 PRELIMINARIES

### 2.1 Unsupervised State Detection for MTS

**Definition 1 (Multivariate Time Series, MTS).** *A multivariate time series (MTS), denoted by $X$, is an ordered sequence of sensory observations:*

$$X = \{x_i\}_{i=1}^{\mathsf{T}}, \quad x_i \in \mathbb{R}^{\mathsf{N}}, \tag{1}$$

*where $x_i$ is the observation at the $i$-th time step; the parameters $\mathsf{N}$ and $\mathsf{T}$ are the MTS dimensionality and the length of the MTS, respectively. A segment of $X \in \mathbb{R}^{\mathsf{N} \times \mathsf{T}}$ spanning time steps $i$ to $j$ is denoted as $X_{i:j} \in \mathbb{R}^{\mathsf{N} \times (j-i)}$.*

**Definition 2 (State in MTS).** *A state acts as a concise representation of the underlying condition associated with an MTS segment.*

States are discernible due to their unique internal features, such as recurring patterns or consistent statistical behaviors [17, 20, 40]. Referring to the dance routine example in Fig. 1, states might correspond to different dance movements, each of which exhibits a unique pattern: "walk" with rhythmic variations, "jump" with intense spikes, "hop" with recurrent bursts, and "run" with higher frequency and intensity than "walk". Accurate identification of these specific patterns (states) is crucial to understanding the underlying process captured by an MTS and to enable downstream applications like urban monitoring [4] and healthcare [35].

**Definition 3 (Unsupervised State Detection, USD).** *Given an MTS $X$ of length $\mathsf{T}$, the process of unsupervised state detection (USD) aims to assign each observation $x_i \in X$ a state index, $s_i$, without any training data and a set of predefined states. This process ultimately yields a state sequence $s = \{s_i\}_{i=1}^{\mathsf{T}}$, where $s_i \in \mathbb{R}^+$ identifies a specific state detected by the USD process.*

Note that the number of distinct states found by the USD process, $\text{Distinct}(s)$, is unknown prior to the start of the process.

We proceed to introduce the **classical USD pipeline** [20, 26, 36]. In general, USD utilizes a *sliding window*, i.e., MTS data is processed by the USD system per window along the time dimension. In Fig. 1, a sliding window of size $\mathsf{P}$ and step size $\mathsf{B}$ traverses the MTS. Let $W_t = X_{t-\mathsf{P}:t}$ be the current window of the MTS. This window is processed by the **MTS encoder** to obtain an embedding $z_t$ in a latent space. This embedding is input to a **clustering model** that deduces its state index $s_t \in \{\mathbb{R}^+\}^{\mathsf{P}}$, which is then assigned to the $\mathsf{P}$ time steps in window $W_t \in \mathbb{R}^{\mathsf{N} \times \mathsf{P}}$.

As the sliding window moves at step size $\mathsf{B}$, each time step eventually has $\lfloor \frac{\mathsf{P}}{\mathsf{B}} \rfloor$ state indexes determined by the USD process[1]. To reconcile these sets of state indexes and ensure smoother transitions, the final state index for each time step is determined through majority voting [20, 26, 36].

---

[1] In general, the initial time steps within the first window and the final time steps within the last window have fewer than $\lfloor \frac{\mathsf{P}}{\mathsf{B}} \rfloor$ associated state indexes.

Conventional DL-based USD methods [26, 36] employ intricate neural networks and take the raw MTS data as input directly. To enhance efficiency, we incorporate a Fast Fourier Transform (FFT)-based time series compressor (see Section 3.1.1). Below, we provide a brief overview of FFT.

## 2.2 FFT in Time Series Analysis

FFT [6] is fundamental to signal processing. Engineered for optimal computational efficiency, FFT calculates the Discrete Fourier Transform of numerical sequences. This capability is essential for identifying frequency components in a time series, enabling noise reduction and compression. Further, this capability supports our goal of reducing the computational overhead of the USD process, since this overhead is correlated with the length of the MTS. Integrating FFT offers a promising avenue for enhancing both efficiency and accuracy.

Specifically, we utilize real-valued FFT. Crafted as a variant of FFT, it transforms an MTS window $W$ with $P$ time steps into $K = (\lfloor P \rfloor/2 + 1)$ frequency components, represented as a matrix $Q \in \mathbb{R}^{N \times K}$. Conversely, the inverse real-valued FFT converts $Q$ back to a new time-domain representation, denoted by $\hat{W}$.

The computation of real-valued FFT and the inverse real-valued FFT are formulated in Equations 2 and 3, respectively.

$$q_k \leftarrow \sum_{p=0}^{P-1} \left( x_p \cos\left(-\frac{2\pi k p}{P}\right) + j \sin\left(-\frac{2\pi k p}{P}\right) \right), k = 0, \ldots, K-1 \quad (2)$$

$$\hat{x}_p \leftarrow \frac{1}{P} \sum_{k=0}^{K-1} \left( q_k \cos\left(\frac{2\pi k p}{P}\right) - q_k j \sin\left(\frac{2\pi k p}{P}\right) \right), p = 0, \ldots, P-1 \quad (3)$$

Here, $q_k$ is the $k$-th ($0 \leq k < K$) frequency component of the real-valued FFT result $Q$, and $\hat{x}_p$ is the $p$-th ($0 \leq p < P$) time step's data of the inverse real-valued FFT result $\hat{W}$.

## 3 KEY TECHNIQUES OF E2USD

E2USD follows the classical USD pipeline (Section 2.1), encompassing MTS embedding and clustering. In particular, E2USD utilizes the Dirichlet Process Gaussian Mixture Model (DPGMM) [5] for clustering. Below, we present the key innovations in E2USD. E2USD first applies a compact embedding procedure to the input MTS to ease subsequent neural computations (Section 3.1). In the compact embedding procedure, E2USD also incorporates a novel False Negative Cancellation Contrastive Learning method (Section 3.2) to ensure or improve the effectiveness of the learned embeddings. Finally, E2USD employs an Adaptive Threshold Detection (ADATD) scheme for improved applicability in online streaming (Section 3.3).

## 3.1 Compact Embedding of MTS

Contemporary DL-based embedding methods often utilize complex end-to-end networks [24, 26, 34, 36], potentially overlooking the rich domain-specific knowledge in traditional feature engineering. To generate concise yet informative embeddings for input MTS, we advocate for revisiting fundamental principles in data analysis and signal processing, particularly in frequency domain analysis [6] and time series decomposition [39]. Our studies indicate that these approaches can yield effective embedding outcomes while using simpler techniques. As shown in Fig. 2, our compact embedding comprises an FFT-based Time Series Compressor (FFTCOMPRESS)

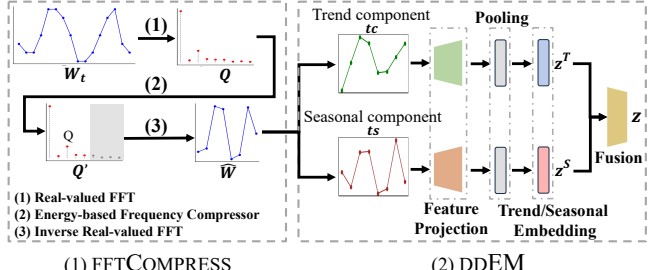

(1) FFTCOMPRESS                    (2) DDEM

**Fig. 2: Compact embedding of the input MTS.**

and a Decomposed Dual-view Embedding Module (DDEM), detailed in Section 3.1.1 and Section 3.1.2, respectively.

*3.1.1 FFTCOMPRESS.* In practical applications, sensor designs often incorporate over-sampling rates to ensure comprehensive information capture. In this sense, maintaining a sparse representation of time series is reasonable and beneficial for retaining essential information and reducing noise, ultimately simplifying subsequent temporal feature extraction. Frequency domain analysis, primarily using FFT [10, 44–46], is a well-established technique to achieve this goal. However, it remains challenging to effectively identify the set of distinct active frequency components for FFT analysis across different tasks. To address this aspect, we introduce FFTCOMPRESS, which encompasses three steps.

**(1) Real-valued FFT**. Given a window $W_t = X_{t-P:t} \in \mathbb{R}^{N \times P}$, this step transforms $W_t$ into its frequency domain representation $Q \in \mathbb{R}^{N \times K}$, using Equation 2. Expressing the input signal in the frequency domain is crucial for our goal of compressing time series data. In the frequency domain, we can identify and prioritize the most significant frequency components, thus achieving compression by emphasizing salient features and discarding less critical ones.

**(2) Energy-based Frequency Compressor**. As the core of FFT-COMPRESS, this step dynamically selects and retains *active frequencies* from the frequency domain based on the cumulative energy observed across all the MTS dimensions. Energy in signal processing quantifies the strength (magnitude) of a signal's frequency components. Typically, low energy implies reduced strength and activation. A possible solution is to select discrete frequencies, rather than forming a continuous band. However, two important factors need to be considered: first, research has shown that noise is often concentrated at the extreme frequencies [46]; second, active frequencies tend to cluster around a central range. Skipping frequencies could potentially lead to the omission of crucial information. Considering these factors, we choose to utilize a continuous frequency band. This decision is especially effective at removing both high- and low-frequency noise, which improves the overall data representation. Specifically, given a frequency-domain representation $Q \in \mathbb{R}^{N \times K}$, we compute for its **cumulative energy** $e \in \mathbb{R}^K$ [31], where the $k$-th component $e_k$ corresponds to a viable starting frequency $k$ ($0 \leq k < K$) and is computed as follows:

$$e_k = \sum_{i=k}^{k+Q-1} \sum_{n=0}^{N-1} (Q_{n,i})^2. \quad (4)$$

Here, $Q$ ($< K$) is a predefined frequency bandwidth; $N$ is dimensionality of the MTS; and $Q_{n,i}$, a scalar, is the amplitude of the $i$-th

frequency component of the $n$-th dimension. We then identify the starting position $k' \in (0, K)$ that yields the maximum cumulative energy over the given bandwidth $Q$. Subsequently, we cut off $Q$ consecutive frequency components starting from the $k'$-th frequency component of $Q$:

$$Q' = Q[0 : N \; ; \; k' : k' + Q]. \quad (5)$$

As a result, $Q'$ encompasses $Q$ consecutive frequency components that capture the most pronounced energy contributions.

**(3) Inverse Real-valued FFT**. The compressed frequency domain representation $Q' \in \mathbb{R}^{N \times Q}$ is then transformed back to its time domain using Equation 3. This yields a compressed MTS $\hat{W} \in \mathbb{R}^{N \times P'}$, where $P' = 2 \times (Q - 1)$. As illustrated in the right part of Fig. 9, for an original time series of length $P = 480$, the energy-based frequency compressor retains a frequency bandwidth of $Q = 41$. This results in a compressed time series of length $P' = 80$. Through FFTCOMPRESS, the MTS is compressed from $P$ to $P'$ in the temporal dimension. This leads to a significant reduction in computational overhead for the subsequent feature extraction while preserving essential signal attributes and reducing noise.

A detailed assessment of the energy-based frequency compressor's impact, along with a comprehensive parameter sensitivity analysis related to $Q$, can be found in the Appendix [1].

*3.1.2 DDEM.* Many DL-based approaches employ computationally intensive modules for MTS feature extraction [24, 26, 34, 36]. While effective, these complex structures may capture redundantly features that can be obtained efficiently using lightweight, traditional tools, thus incurring unnecessary computational costs. Time series decomposition is a widely used technique for extracting essential components like trend and seasonality. Recognizing its significance, we introduce the Decomposed Dual-view Embedding Module (DDEM), which features an innovative and lightweight architecture that seamlessly combines time series decomposition with a subsequent lightweight dual-view neural embedding module. It facilitates the embedding of both trend and seasonality in MTS by leveraging the strengths of both traditional and modern approaches.

**(1) Decomposition of Compressed MTS**. Studies [13, 38, 42] show that time series data can be broken down into trend, seasonal, and residual (noise) components. In our approach, we exclude the residual component, as the prior FFTCOMPRESS step has eliminated noise. Thus, with the compressed MTS $\hat{w} \in \mathbb{R}^{N \times P'}$, we employ a proven method, the *moving average* scheme [42], for decomposing it into trend and seasonal components. This approach is well recognized for its effectiveness in this context.

- The **trend component** $tc \in \mathbb{R}^{N \times P'}$ is calculated using a moving average kernel of size $\kappa$, which is odd, as follows:

$$tc[n, t] = \frac{1}{\kappa} \sum_{i=-(\kappa-1)/2}^{(\kappa-1)/2} \hat{w}[n, t + i]. \quad (6)$$

- The **seasonal component** $sc \in \mathbb{R}^{N \times P'}$ is obtained by subtracting the trend component $tc$ from $\hat{w}$, formally, $sc = \hat{w} - tc$.

**(2) Dual-view Embedding**. Referring to Fig. 2, after decomposing the signals into two distinct views, they undergo embedding using lightweight networks. Both trend and seasonal views share identical embedding structures (1D convolution + max pooling + linear

embedding). The embeddings resulting from both views are fused to create a compact MTS embedding.

Feature Projection Layer (no training). The trend and seasonal components are projected into high-dimensional latent spaces using 1D convolution:

$$h^T = \text{Conv1D}(tc) \quad h^S = \text{Conv1D}(sc). \quad (7)$$

This step employs convolution feature maps to provide varying perspectives on the signals. Notably, this projection module does not require training. It has been proven effective in multiple MTS classification studies [14, 15, 30]. The subsequent layers are trained to extract features from each view. Additionally, a detailed parameter sensitivity analysis regarding the dimensionality of these two latent spaces can be found in the Appendix [1].

Linear Embedding Layer (trainable). A max pooling layer is used to reduce dimensionality, thus enhancing computational efficiency while highlighting the most informative features. Then, a linear layer is applied to effectuate the embedding:

$$z^T = \text{ReLU}(\text{Linear}(\text{MaxPooling1D}(h^T); \Theta^T)), \quad (8)$$

$$z^S = \text{ReLU}(\text{Linear}(\text{MaxPooling1D}(h^S); \Theta^S)), \quad (9)$$

where $z^T \in \mathbb{R}^D$ and $z^S \in \mathbb{R}^D$ are the trend and seasonal embeddings of size $D$, respectively. Trainable parameters $\Theta^T$ and $\Theta^S$ for $\text{Linear}(\cdot)$ encompass weights and biases.

Fusion Layer (trainable). The embeddings from both views are concatenated and further transformed to generate the final MTS embedding $z \in \mathbb{R}^D$, providing a comprehensive representation that captures inter-view relationships:

$$z = \text{Linear}(\text{Concat}(z^T, z^S); \Theta), \quad (10)$$

where $\Theta$ is the corresponding trainable parameters. For a detailed sensitivity analysis regarding the final embedding size $D$, refer to the Appendix [1].

By employing FFTCOMPRESS and DDEM, we avoid intricate neural networks and their computational demands. In the following section, we introduce an innovative contrastive learning scheme to ensure that the compact embedding structure preserves crucial information effectively.

## 3.2 False Negative Cancellation Contrastive Learning for Effective Embedding

As a technique used widely in the embedding stage of USD, contrastive learning [11] maximizes the similarity between similar samples (so-called positive pairs), while minimizing it for dissimilar ones (so-called negative pairs). However, **false negative sampling** is a common issue: current approaches [17, 34, 36] involve randomly sampling $U$ distinct window groups from an MTS, each having $V$ consecutive windows (see Fig. 3) through a sliding window. With this setup, each group is assumed to represent a unique state, with positive sample pairs always being from the same group and negative sample pairs always being from different groups. However, this setup can lead to a problem, where different groups inadvertently share the same state, causing samples from these groups to be incorrectly regarded as negative pairs. In Fig. 3, the blue and green windowed samples, which come from different groups, are

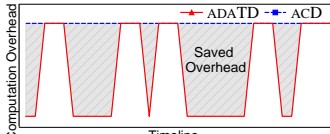

**Fig. 3: Overview of $\mathcal{L}_{\text{FNCC}}$. Same-color windows belong to the same group. The blue and green groups denote a pair of *false negatives*; these are random selections labeled as negative pairs, although they share the same state.**

incorrectly classified as negative pairs. In reality, they belong to the same state "walk" and should be considered as positive pairs.

We thus propose FNCCLEARNING (False Negative Cancellation Contrastive Learning), a novel approach that addresses this issue with a unique similarity-based negative sampling scheme. By considering seasonal and trend embeddings, it can sample those genuinely dissimilar *negative pairs* from the groups with low similarity, enhancing MTS embedding effectiveness for USD. We proceed to present the new sampling scheme, followed by the overall FNC-CLEARNING loss.

**(1) Similarity-based Negative Sampling.** This scheme ensures the selection of genuinely dissimilar negative pairs via a similarity-based approach, evaluating seasonal and trend similarities for each pair and retaining the least similar pairs. The process involves three main steps:

Listing Possible Negative Pairs. Let $\cup$ be the number of randomly sampled window groups, each of which contains $\vee$ consecutive windows. A comprehensive set, $\mathcal{J}$, is compiled encompassing all conceivable pair combinations from a total of $\cup$ groups, resulting in $\cup \times (\cup - 1)/2$ possible negative pairs.

Computing Similarities for Each Pair. For each pair $(i, j) \in \mathcal{J}$, we compute seasonal ($\text{sim}_{i,j}^{S}$) and trend ($\text{sim}_{i,j}^{T}$) similarities using dot products, capturing both seasonal patterns and evolving trends:

$$\text{sim}_{i,j}^{S} = (\boldsymbol{d}_i^S)^\top \cdot \boldsymbol{d}_j^S; \quad \text{sim}_{i,j}^{T} = (\boldsymbol{d}_i^T)^\top \cdot \boldsymbol{d}_j^T, \quad (11)$$

where $\boldsymbol{d}_i^S$ (*resp.* $\boldsymbol{d}_i^T$) represent the centroid (i.e., average embedding) of the seasonal embeddings $\boldsymbol{z}^S$ (*resp.* trend embeddings $\boldsymbol{z}^T$) of all $\vee$ consecutive windowed samples in the $i$-th window group.

The comprehensive similarity, $\text{sim}_{i,j}^{O}$, for each pair $(i, j)$ is finally computed as the product of the trend and seasonal similarities:

$$\text{sim}_{i,j}^{O} = \text{sim}_{i,j}^{T} \cdot \text{sim}_{i,j}^{S} \quad (12)$$

Filtering True False Negative Pairs. The $\lfloor \lambda \times |\mathcal{J}| \rfloor$ least similar pairs based on $\text{sim}_{i,j}^{O}$, with $\lambda$ as a fraction parameter in $(0, 1]$ (set to 0.5 by default), are selected to form the set $\mathcal{J}'$ of negative pairs.

**(2) FNCCLEARNING Loss.** The following negative loss $\mathcal{L}_{\text{neg}}$ aims to minimize the similarity between negative pairs from $\mathcal{J}'$:

$$\mathcal{L}_{\text{neg}} = \frac{1}{|\mathcal{J}'|} \sum_{(i,j) \in \mathcal{J}'} -\log(\text{Sigmoid}(-\boldsymbol{d}_i^\top \cdot \boldsymbol{d}_j)), \quad (13)$$

where $\boldsymbol{d}_i$ refers to the centroid of the final embedding $\boldsymbol{z}$ of all $\vee$ consecutive windowed samples in the $i$-th window group.

In Fig. 3, it is assumed that consecutive windowed samples from the same group (indicated by the same color) share a uniform state.

Accordingly, the positive loss $\mathcal{L}_{\text{pos}}$ aims to maximize the similarity between their embeddings:

$$\mathcal{L}_{\text{pos}} = 1/\mathsf{M} \sum_{k=0}^{\cup-1} \sum_{i=0}^{\vee-1} \sum_{j=0,j<i}^{\vee-1} -\log(\text{Sigmoid}((\boldsymbol{z}_{k,i})^\top \cdot (\boldsymbol{z}_{k,j}))), \quad (14)$$

where $\boldsymbol{z}_{k,i}$ represents the embedding of the $i$-th ($0 \le i < \vee$) window from the $k$-th ($0 \le k < \cup$) group, and $\mathsf{M} = \frac{\cup \times \vee \times (\vee-1)}{2}$ is used for normalization.

Ultimately, the FNCCLEARNING aims to enhance the similarity among positive pairs while diminishing the similarity among negative pairs. Thus, the FNCCLEARNING loss is defined as the sum of its positive and negative loss components:

$$\mathcal{L}_{\text{FNCC}} = \mathcal{L}_{\text{pos}} + \mathcal{L}_{\text{neg}} \quad (15)$$

The two hyperparameters, $\cup$ and $\vee$, directly impact the loss computation, and the evaluation of their impact has been provided in the Appendix [1].

### 3.3 Streaming USD with Adaptive Threshold

**Fig. 4: Saved overhead.**

Upon obtaining the embedding of each windowed sample, a clustering algorithm, typically the Dirichlet Process Gaussian Mixture Model (DPGMM) [5], is traditionally employed. However, current methodologies often do not take into account the challenges of real-world online streaming. In such scenarios, states tend to persist, making clustering unnecessary for successive samples. To avoid redundant clustering, we propose the Adaptive Threshold Detection (ADATD) mechanism that defers clustering until a new sample shows low similarity (controlled by an adaptive threshold) to the previous one. This ensures that clustering is invoked only when needed, enhancing USD efficiency. Fig. 4 illustrates ADATD's computational savings compared to the classical "Always Clustering Detection" (ACD) mechanism.

The ADATD process is detailed in Algorithm 1. Specifically, to gauge the temporal consistency between the current sample and its predecessor, we employ the dot product to quantify the similarity between their respective embeddings, $\boldsymbol{z}_{\text{pre}}$ and $\boldsymbol{z}_t$:

$$\text{sim}(\boldsymbol{z}_{\text{pre}}, \boldsymbol{z}_t) = \boldsymbol{z}_{\text{pre}}^\top \cdot \boldsymbol{z}_t. \quad (16)$$

Then, this similarity value is compared to an adaptive threshold $\tau$. If the similarity is below this threshold, this indicates a likely state transition, triggering clustering to identify the new state $s_t$.

$$s_t \leftarrow \begin{cases} s_{\text{pre}} & \text{if } \text{sim}(\boldsymbol{z}_{\text{pre}}, \boldsymbol{z}_t) \ge \tau \\ \text{Cluster}(\boldsymbol{z}_t) & \text{otherwise} \end{cases} \quad (17)$$

The overall ADATD process is outlined in Algorithm 1, where the threshold $\tau$ adapts to the context of streaming MTS. Its tuning is controlled by the computed similarity with two *scaling factors* $\delta_i$ and $\delta_r$. When the similarity exceeds $\tau$, it implies that the state remains unchanged, prompting an increase in the threshold by the scaling factor $\delta_i$ (line 10 in Algorithm 1), as the growing probability of a state transition. Conversely, if the similarity falls below the threshold, we hypothesize a state transition, prompting a clustering

operation for confirmation (line 12). Upon verification (line 13), $\tau$ is raised to act conservatively against the state transition (line 15). If the transition is deemed false, the threshold is decreased to counter overestimation from an excessively high threshold (line 16), and a higher value of the scaling factor $\delta_r$ is selected to ensure a rapid response to incorrect state transition hypotheses. Concurrently, a reduced value for $\delta_i$ is preferred, particularly considering the probability of an impending state transition rises. Parameter sensitivity study of $\delta_r$ and $\delta_i$ is reported in the Appendix [1].

In general, ADATD seamlessly incorporates a cost-effective similarity metric tailored for streaming USD, significantly reducing redundant clustering operations. Empirical evaluation in Section 4.4 confirms the efficacy of ADATD.

---

**Algorithm 1** Adaptive Threshold Detection (ADATD)

---

**Require:** MTS stream $X$, threshold $\tau$, and scaling factors $\delta_r$ and $\delta_i$
**Ensure:** State $s_t$ for continuous time step $t$ on $X$

  1: $W_0 \leftarrow X_{0:P-1}$                    ▷ sliding window sampling
  2: $z_0 \leftarrow \text{CompactEmbedding}(W_0)$         ▷ see Section 3.1
  3: $s_0 \leftarrow \text{Clustering}(z_0)$                 ▷ DPGMM
  4: $z_{\text{pre}}, s_{\text{pre}} \leftarrow z_0, s_0$             ▷ initialize state
  5: **while** obtaining updated $x_t$ from $X$ **do**
  6:     $W_t \leftarrow X_{t-P+1:t}$             ▷ sliding window sampling
  7:     $z_t \leftarrow \text{CompactEmbedding}(W_t)$
  8:     **if** $\text{sim}(z_{\text{pre}}, z_t) \geq \tau$ **then**
  9:         $s_t \leftarrow s_{\text{pre}}$             ▷ keep current state
10:         $\tau \leftarrow \tau \times (1 + \delta_i)$          ▷ increase $\tau$
11:     **else**
12:         $s_t \leftarrow \text{Clustering}(z_t)$       ▷ acquire new state
13:         **if** $s_t \neq s_{\text{pre}}$ **then**     ▷ verify state transition
14:             $z_{\text{pre}}, s_{\text{pre}} \leftarrow z_t, s_t$      ▷ update state
15:             $\tau \leftarrow \tau \times (1 + \delta_i)$      ▷ increase $\tau$
16:         **else** $\tau \leftarrow \tau \times (1 - \delta_r)$      ▷ decrease $\tau$

---

## 4 EXPERIMENTS

### 4.1 Experimental Settings

The entire codebase, datasets, hyperparameter settings, and instructions are available at http://bit.ly/3rMFJVv. We trained the DL models on a server with an NVIDIA Quadro RTX 8000 GPU. For model inference, we employed an Intel Xeon Gold 5215 CPU (2.50GHz). Additionally, we carried out a case study of MCU deployment using an STM32H747 device [2]. Further implementation details can be found in the Appendix [1].

**Baselines**. The following baselines are introduced. Baselines 1–3 employ the USD pipeline outlined in Section 2.1, while the remaining ones do not. **(1)** HVGH [26] employs a variational autoencoder for encoding MTS windows and utilizes the Hierarchical Dirichlet Process (HDP) for clustering. **(2)** TICC [20] uses a correlation network for encoding and adopts Toeplitz inverse covariance-based clustering. **(3)** TIME2STATE [36] employs a Temporal Convolutional Network to encode MTS windows and utilizes the DPGMM for clustering (as does E2USD). **(4)** AUTOPLAIT [25] applies the Minimum Description Length principle to segment the MTS and recursively models each segment with the Hidden Markov Model. **(5)** CLASPTS_KMEANS [16] identifies change points in an MTS using

multiple binary classifiers and employs KMeans [32] for segment clustering. **(6)** HDP_HSMM [33] is a Bayesian non-parametric extension of the Hidden Semi-Markov Model that uses HDP to estimate the number of states.

**Datasets**. For evaluations, we employ six datasets used in previous studies [25, 36]. These include one synthetic dataset, Synthetic [36], and five real-world datasets from diverse fields: MoCap, ActRecTut, PAMAP2, and UscHad track various human activities [7, 25, 28, 43], and UcrSeg covers MTS from applications such as insect research, robotics, and energy [18]. Among these, PAMAP2 exhibits the largest MTS lengths (ranging from 253k to 408k), while UcrSeg has the shortest (varying between 2k and 40k). UscHad features the largest number of states (12 in total), whereas UcrSeg has the fewest, with up to 3 states. Notably, UcrSeg stands out as a univariate times series dataset. A detailed description of the datasets is available in the Appendix [1].

**Metrics**. We use the Adjusted Rand Index (ARI) and Normalized Mutual Information (NMI) to assess the detection accuracy, as does prior research [36]. ARI quantifies the instance-wise consistency between predicted and ground truth clusters by emphasizing clustering granularity, while NMI measures the shared information between ground truth and model clusterings. Furthermore, for evaluating detection efficiency, we present the Processing Time (PT), which records the time in seconds required by each method to process USD over a specific length of an MTS.

### 4.2 Overall Comparison

An overall comparison between E2USD and the baselines across the six datasets is listed in Table 1. The comparison is conducted on an Intel CPU over the STM device due to higher computational demands for the baselines that surpass the STM device's capacity. E2USD consistently exhibits top-tier performance, achieving the best or near-best ARI and NMI scores. Notably, E2USD also exhibits superior efficiency, as evidenced by consistently having the shortest processing time (PT)[2]. Among the baselines, TIME2STATE manifests high performance at ARI, NMI, and PT for most datasets, positioning itself as the strongest competitor to E2USD. Later in this section, a more detailed comparison between E2USD and TIME2STATE is provided. AUTOPLAIT excels in the MoCap dataset—notable for its shortest average MTS length—achieving the highest ARI and NMI scores, albeit with longer PT. However, it struggles on other datasets, particularly as it is unable to process the PAMAP2 dataset (marked as 'N/A' in Table 1), which has the longest average MTS length. CLASPTS_KMEANS achieves the top ARI and NMI scores on the univariate UcrSeg dataset. However, its performance is not competitive on other datasets of MTS.

Efficiency is crucial in real-world applications, especially with large sequences or variable data volumes. Thus, we conduct experiments over sequence lengths ranging from 40k to 400k, using the Synthetic dataset. As illustrated in Fig. 5 (a), E2USD exhibits by far the lowest overall processing times throughout the tested range of sequence lengths, from 0.52s for 40k to 1.71s for 400k. This renders E2USD ideally suited for real-world scenarios demanding swift responses and the capacity to adapt to varying data volumes.

---

[2]Here, PT denotes the processing time in seconds for each method applied to a medium-sized MTS (40k length) from the given dataset.

**Table 1: Overall comparison.**

| Method | Synthetic | | | MoCap | | | ActRecTut | | | PAMAP2 | | | UscHad | | | UcrSeg | | |
|---|---|---|---|---|---|---|---|---|---|---|---|---|---|---|---|---|---|---|
| | PT | ARI | NMI | PT | ARI | NMI | PT | ARI | NMI | PT | ARI | NMI | PT | ARI | NMI | PT | ARI | NMI |
| HVGH [26] | 27.53 | 0.0809 | 0.1606 | 25.98 | 0.0500 | 0.1523 | 26.17 | 0.0881 | 0.2088 | 24.06 | 0.0032 | 0.0374 | 25.42 | 0.0788 | 0.1883 | 26.74 | 0.0638 | 0.1451 |
| HDP_HSMM [33] | 51.24 | 0.6619 | 0.7798 | 55.48 | 0.5509 | 0.7230 | 56.57 | 0.6644 | 0.6473 | 52.10 | 0.2882 | 0.5338 | 53.43 | 0.4678 | 0.6839 | 49.38 | 0.1625 | 0.2574 |
| TICC [20] | 20.55 | 0.6242 | 0.7489 | 21.09 | 0.7218 | 0.7524 | 22.41 | 0.7839 | 0.7466 | 24.94 | 0.3008 | 0.5955 | 21.51 | 0.3947 | 0.7028 | 19.29 | 0.2325 | 0.2158 |
| Aᴜᴛᴏᴘʟᴀɪᴛ [25] | 73.80 | 0.0713 | 0.1307 | 76.59 | 0.8057 | 0.8289 | 257.94 | 0.0586 | 0.1418 | N/A | N/A | N/A | 103.09 | 0.2948 | 0.5413 | 19.48 | 0.0688 | 0.1035 |
| CʟᴀSPTS_KMᴇᴀɴs [16] | 33.34 | 0.2950 | 0.4480 | 35.17 | 0.5450 | 0.6763 | 91.59 | 0.2825 | 0.2309 | 74.41 | 0.1700 | 0.5830 | 48.01 | 0.5075 | 0.6940 | 8.44 | 0.5050 | 0.5035 |
| Tɪᴍᴇ2Sᴛᴀᴛᴇ [36] | 2.30 | 0.8176 | 0.8407 | 2.37 | 0.7529 | 0.7584 | 2.85 | 0.7670 | 0.7407 | 2.66 | 0.3135 | 0.5905 | 2.44 | 0.6522 | 0.8126 | 2.19 | 0.4325 | 0.4429 |
| E2Usᴅ (Ours) | 0.63 | 0.8843 | 0.8025 | 0.66 | 0.7896 | 0.7812 | 0.72 | 0.7909 | 0.7473 | 0.79 | 0.3345 | 0.6143 | 0.68 | 0.6833 | 0.8164 | 0.60 | 0.3678 | 0.4468 |

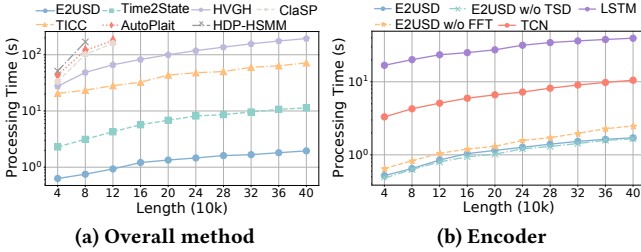

**(a) Overall method**      **(b) Encoder**

**Fig. 5: Efficiency comparison.**

**Table 2: Key metrics comparison: Tɪᴍᴇ2Sᴛᴀᴛᴇ vs. E2Usᴅ.**

| Metric | Tɪᴍᴇ2Sᴛᴀᴛᴇ | E2Usᴅ | Acceleration |
|---|---|---|---|
| Total Params | 82,218 | **2,764** | 29.7 × |
| Trainable Params | 82,218 | **684** | 120.2 × |
| MACC (M) | 5.01 | **0.06** | 83.5 × |
| Peak Memory (MB) | 11.44 | **0.05** | 228.8 × |

**E2Usᴅ vs. Tɪᴍᴇ2Sᴛᴀᴛᴇ**. We compare with the current SOTA model Tɪᴍᴇ2Sᴛᴀᴛᴇ on the Synthetic dataset. Both methods utilize DL-based encoders and share the same clustering model, directing our focus primarily on the encoder component. As depicted in Table 2, E2Usᴅ outperforms Tɪᴍᴇ2Sᴛᴀᴛᴇ significantly in terms of computational and storage efficiency. To be precise, E2Usᴅ requires roughly 30 times fewer total parameters, 120 times fewer trainable parameters, and reduces Multiply-ACCumulate (MACC) counts[3] by an impressive 83.5 times. Moreover, its peak memory usage is about a factor of 229 times smaller. All in all, these statistics provide evidence of E2Usᴅ's strength in resource-constrained scenarios.

## 4.3 Component Study of E2Usᴅ

This section evaluates the effectiveness and efficiency of E2Usᴅ's proposed components. As shown in Section 4.2, E2Usᴅ exhibits relatively high consistency between ARI and NMI scores. Due to space limit, we thus focus on reporting the ARI scores. The corresponding NMI results are available in the Appendix [1].

*4.3.1 Encoder.* We compare our encoder (denoted as E2Usᴅ) with two variants: one without ꜰꜰᴛCᴏᴍᴘʀᴇss (E2Usᴅ w/o FFT) and the other without Trend-Seasonal Decomposition of ᴅᴅEM (E2Usᴅ w/o TSD). Besides, we include widely used MTS encoders LSTM [17] and TCN [36] for comparison. To maintain fairness, we only substitute the encoder component of E2Usᴅ with these alternatives, while keeping all the other settings unchanged.

---

[3]The MACC count refers to the aggregate of multiply-accumulate operations in a given algorithm, commonly used as a metric for computational complexity in DL.

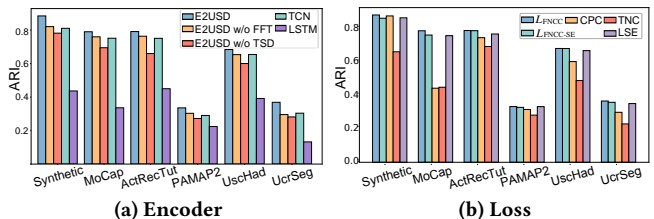

**(a) Encoder**      **(b) Loss**

**Fig. 6: Effectiveness comparison for component study.**

We first examine the efficiency of all encoders using the setting described in Section 4.2. Fig. 5 (b) reveals that E2Usᴅ consistently offers top-tier temporal efficiency, starting at a 0.52s PT for a 40k sequence and only slightly increasing to 1.71s for a 400k sequence. When comparing E2Usᴅ to its variants, we observe that introducing TSD marginally increases PT but significantly boosts accuracy (explained later). Conversely, integrating ꜰꜰᴛCᴏᴍᴘʀᴇss leads to a notable reduction in PT. Further, LSTM and TCN increase the processing time substantially, with LSTM at 39.46s and TCN at 10.51s for processing a 400k sequence.

Referring to the accuracy results reported in Fig. 6 (a), E2Usᴅ consistently outperforms its competitors across all datasets. Notably, the inclusion of ꜰꜰᴛCᴏᴍᴘʀᴇss does not compromise accuracy, owing to its noise reduction capability, while TSD significantly enhances accuracy (see E2Usᴅ vs. E2Usᴅ w/o TSD). When juxtaposed with LSTM and TCN, E2Usᴅ also demonstrates better performance. One of the distinct advantages of E2Usᴅ is its ability to clearly extract valid frequency and period trend information, which is crucial for accurate clustering. While LSTM and TCN have their merits, their black-box nature makes it uncertain whether they can effectively capture this information as reliably as the trend-seasonal decomposition feature of E2Usᴅ.

*4.3.2 ꜰɴᴄᴄLᴇᴀʀɴɪɴɢ Loss.* We compare our proposed $\mathcal{L}_{\text{FNCC}}$ with SOTA loss functions, including Temporal Neighborhood Coding (TNC) [34], Contrastive Predictive Coding (CPC) [24], and Latent State Encoding (LSE) [36]. Besides, we include a variant of $\mathcal{L}_{\text{FNCC}}$, $\mathcal{L}_{\text{FNCC-SE}}$, by constructing negative pairs using Samples' Embeddings (SE), rather than employing the centroids (i.e., average embeddings) of these groups. As emphasized in existing studies, these losses are encoder-agnostic [24, 34, 36]. To ensure fairness, we only replace the loss function of E2Usᴅ, following established research conventions [36]. This setup ensures that any performance differences stem solely from the inherent qualities of the loss functions themselves, rather than variations in encoder architectures.

Fig. 6 (b) shows that $\mathcal{L}_{\text{FNCC}}$ consistently outperforms baselines on all datasets.A key factor contributing to this robust performance

is $\mathcal{L}_{\text{FNCC}}$'s effectiveness in minimizing false negatives, which leads to a cluster-friendly latent embedding space and enhances accuracy. Note that incorporating $\mathcal{L}_{\text{FNCC}}$ does not compromise the efficiency, as trained DL models remain loss-agnostic.

## 4.4 Efficacy of ADATD

We have empirically evaluated the performance of the Adaptive Threshold Detection (ADATD) algorithm when applied to the processing of streaming MTS data using the Synthetic dataset. We simulate streaming scenarios by continuously feeding MTS data to the model. Our assessment involved a comparison with two baseline detection schemes: "Always Clustering Detection" (ACD) and "Static Threshold Detection" (sTD), with sTD($\tau$) representing detection based on a fixed threshold value $\tau$.

As depicted in Fig. 7 (a) and (b), ADATD demonstrates a well-balanced trade-off between accuracy and efficiency. While ACD achieves slightly higher accuracy, ADATD excels significantly in terms of efficiency. Furthermore, ADATD surpasses sTD across a range of static thresholds in terms of accuracy while requiring fewer clustering operations and maintaining competitive processing time. Notably, when increasing the static thresholds in sTD (i.e., $\tau$ ranging from 0.2 to 0.8), the detection accuracy improves and approaches that of ADATD at $\tau = 0.8$, However, this increase in $\tau$ is accompanied by a decrease in efficiency, and even at $\tau = 0.4$, it still falls short of matching the efficiency of ADATD. This observation highlights the superior adaptability of ADATD to variations in the similarity across different states, thus ensuring efficient and accurate detection.

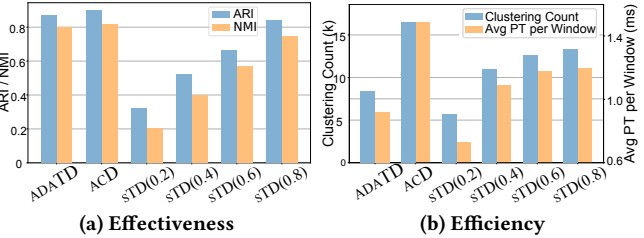

(a) Effectiveness          (b) Efficiency

Fig. 7: Comparative analysis of ADATD with ACD and sTD.

## 4.5 Case Study on Resource-limited MCU

To assess the viability of deploying E2USD on edge devices, we conducted experiments using a commodity STM32H747 MCU on the Synthetic dataset. Significantly, this device could *not* accommodate other baseline methods due to their high demands on computation and memory. The results reveal the following operational metrics during the operational phase. The Flash memory consumption amounts to **63.72 KB**, a mere 3.11% of the available 2 MB. In terms of RAM, it uses **73.27 KB**, representing a modest 7.16% of the overall 1 MB capacity, signifying efficient memory utilization. Moreover, the latency for processing each sample is **44.95 ms**, which equates to a detection frequency close to 20 Hz. When assuming a state persists for 10 sampling intervals, E2USD is capable of handling streaming USD scenarios below 200 Hz, encompassing a wide range of practical applications. These results constitute strong evidence of the efficient resource utilization of E2USD, underscoring its applicability in resource-limited scenarios.

## 5 RELATED WORK

**Unsupervised State Detection for MTS.** Broadly, USD for MTS can be categorized into two groups: those that follow the two-stage pipeline outlined in Section 2.1 and those that deviate from it. Research in the former category typically places its focus on the MTS embedding stage. A notable example is HVGH [26], which employs a variational autoencoder for MTS encoding and the Hierarchical Dirichlet Process for clustering. Besides, TICC [20] proposes a novel correlation network based on Toeplitz inverse covariance for MTS embedding. Recently, TIME2STATE [36] introduced contrastive learning to enhance the learning of the embedding module, but it faces challenges of computational overhead and false negative samples, highlighting the need for efficient models for resource-constrained devices like MCUs. In contrast to the two-stage pipeline, methods like AUTOPLAIT [25], HDP_HSMM [33], and CLASPTS_KMEANS [16] adhere to a one-stage framework but encounter significant scalability and stability issues (see Fig. 5 (a)), rendering them unsuitable for online usage.

**Compact Unsupervised Representation Learning for MTS.** While numerous DL studies have explored unsupervised representation learning for MTS data, the majority of current research has concentrated on innovating intricate structures to enhance representation effectiveness [16, 20, 26, 36] but has not considered developing compact models for this purpose. There are also studies dedicated to compact DL models for MTS, offering techniques that can be adapted for unsupervised MTS representation learning. For example, the recent LIGHTCTS [23] introduces compact architectures and operators for MTS forecasting. Similarly, LIGHTTS [8] employs adaptive ensemble distillation to achieve a compact architecture for MTS classification. However, these studies tend to exclusively explore DL approaches, overlooking the traditional methods that have been developed over the years, which often exhibit a higher level of compactness compared to DL structures.

Recognizing this gap, E2USD aims to take into account the nature of MTS data, bridging traditional MTS representation techniques and DL methods. The method leverages an FFT-based approach to obtain a sparse representation of MTS data and then employs a decomposed dual-view embedding module, which integrates classical time series decomposition and a lightweight DL model to produce the final embedding. This offers a promising avenue for compact unsupervised MTS representation learning.

## 6 CONCLUSION AND FUTURE WORK

In this study, we present E2USD, an efficient-yet-effective method for unsupervised MTS state detection. An extensive empirical study offers detailed insight into the properties of the components of E2USD, including FFTCOMPRESS, DDEM, and FNCCLEARNING. Overall, E2USD achieves state-of-the-art accuracy and efficiency in diverse scenarios. The incorporation of an Adaptive Threshold Detection (ADATD) enables a harmonious balance between accuracy and computational requirements, positioning E2USD as the best choice for streaming state detection. As we move forward, we aim to investigate the false positive cases in FNCCLEARNING and explore wider real-world applications for E2USD.

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

# 7 APPENDIX

## 7.1 Dataset Description

We use five real-world datasets and one synthetic dataset for comprehensive evaluations:

- Synthetic [36]: It is a synthetic dataset, generated by the MTS generator TSAGen [37].
- MoCap [25]: Derived from the CMU motion capture repository. In this dataset, every motion is represented as a sequence of hundreds of frames.
- ActRecTut [7]: This dataset involves two participants performing hand movements with height gestures in daily life and 3D gestures while playing tennis.
- PAMAP2 [28]: Covering both basic (e.g., walking, sitting) and composite human activities (e.g., soccer), this dataset features data from eight individuals.
- UscHad [43]: This dataset encompasses 12 distinct human activities such as jumping and running, recorded for 14 individuals.
- UcrSeg [18]: This dataset encompasses diverse sources, including medical fields, insect studies, robotics, and power demand data.

A summary of the statistics is provided in Table 3. Specifically, the varying range denoted as $x - y$ for the number of states, i.e., #(State), implies that an individual time series within the MTS can encompass as few as $x$ states and as many as $y$ states. The same applies to the length and state duration.

**Table 3: Statistics of the datasets.**

| Dataset | #(MTS) | #(State) | #(Variate) | Length (k) | State Duration (k)* |
|---|---|---|---|---|---|
| Synthetic | 100 | 5 | 4 | 9.3-23.7 | 0.1-3.9 |
| MoCap | 9 | 5-8 | 4 | 4.6-10.6 | 0.4-2.0 |
| ActRecTut | 2 | 6 | 10 | 31.4-32.6 | 0.02-5.1 |
| PAMAP2 | 10 | 11 | 9 | 253-408 | 2.0-40.3 |
| UscHad | 70 | 12 | 6 | 25.4-56.3 | 0.6-13.5 |
| UcrSeg | 32 | 2-3 | 1 | 2-40 | 1-25 |

*The state duration is the range of continuous length of a state based on ground truth.

## 7.2 Implementation Details

Experiments were conducted on a server with an NVIDIA Quadro RTX 8000 GPU and an Intel Xeon Gold 5215 CPU (2.50GHz). For FFTCOMPRESS in Section 3.1.1, the frequency bandwidth Q is set to 33 (see Equation (4) and Equation (5)). For DDEM in Section 3.1.2, $\kappa$ in Equation (6) is set to 5, the dimension of the intermediate embeddings $h^{\mathrm{T}}$ and $h^{\mathrm{S}}$ (see Equation (7)), denoted as C, is set to 80 by default, the random convolution kernel for Conv1D in Equation (7) is set to 3, and the dimension of the final embedding $z$ in Equation (7), denoted as D is set to 4. The FNCCLEARNING method used U = 20 groups of windows (each with V = 4 neighboring windows) and a fraction threshold $\lambda$ of 0.5. Sliding window sizes were 128, 256, or 512, dataset-dependent, with step size B = 50. Adam optimizer [22] was used with a learning rate of 0.003 for 20 epochs. Lastly, ADATD was configured with scaling factors $\delta_i = 0.08$ and $\delta_r = 0.1$ and initiated the threshold $\tau$ at a value of 1. The sensitivity of various key parameters, including U, V, Q, D, C, $\delta_i$, and $\delta_r$ are reported later in this appendix.

The MCU deployment uses an STM32H747 device [2] with a 480 MHz Arm Cortex-M7 core, 2 MB Flash memory, and 1 MB RAM, as presented in Fig. 8. The E2USD model was converted to ONNX format and translated to C code with X-CUBE-AI [3]. The C code

was compiled using an ARM-specific version of GCC to create an executable binary.

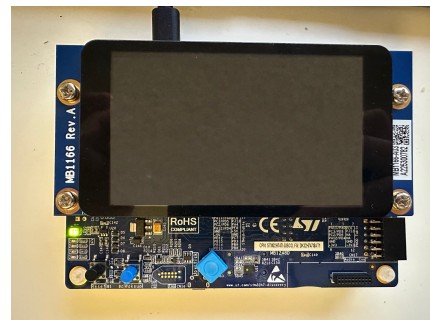

**Fig. 8: The STM32H747 device for model deployment.**

## 7.3 Impact Assessment of the Energy-based Frequency Compressor

To verify the effectiveness of the Energy-based Frequency Compressor (EFC), we conduct FFTCOMPRESS on the UcrSeg dataset using various bandwidth values, denoted as Q = [120, 60, 40]. Referring to Fig. 9, the top row showcases the original and reconstructed waveforms within the native time domain. The middle row displays their corresponding amplitude spectra, while the bottom row exhibits the compressed waveforms. The original data is represented in blue, the reconstructed versions in orange, and the compressed versions in green.

Upon reverting the filtered frequency components to the original time domain, we observe *minimal distortion*. Specifically, the Mean Absolute Percentage Error (MAPE) is less than 5%, even though we retain only a sixth of the original frequency domain representation.

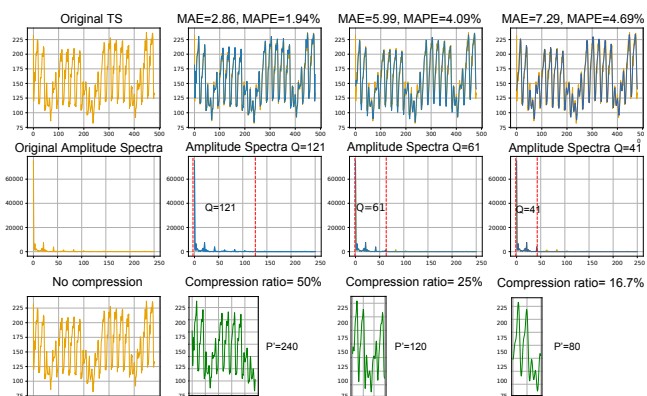

**Fig. 9: Impact assessment of the Energy-based Frequency Compressor on the performance of FFTCOMPRESS.**

## 7.4 Additional NMI Results for Component Study

In Fig. 10, we present the NMI comparisons for both encoders and losses. We note that the trends in NMIs are basically consistent with the corresponding ARIs shown in Fig. 6. This observation further substantiates the effectiveness of our proposed encoder and loss components.

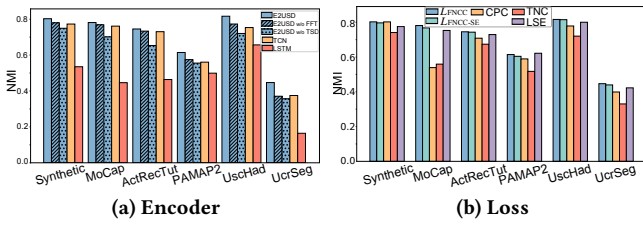

(a) Encoder                    (b) Loss

**Fig. 10: NMI comparison.**

## 7.5 Parameter Sensitivity Study

We conduct a comprehensive parameter analysis using the ActRec-Tut dataset, focusing on assessing how these key parameters affect the ARI and NMI.

*7.5.1 Impact of* U *and* V *in Negative Sampling.* These two parameters jointly contribute to the computation of $\mathcal{L}_{\text{FNCC}}$. More specifically, U designates the number of distinct window groups, whereas V defines the number of consecutive windows within each group. As shown in Fig. 11, altering V does not consistently influence ARI. This unexpected result could be attributed to larger V values capturing broader temporal scopes, thereby introducing false positives due to state transitions. This implies that while increasing V appears to be beneficial for capturing more data, it may inadvertently degrade performance. Conversely, ARI remains stable across a range of U values, highlighting the robustness of E2Usᴅ.

*7.5.2 Impact of Frequency Bandwidth* Q *in* ғғтCᴏᴍᴘʀᴇss. This parameter Q serves as the size of the frequency bandwidth of Energy-based Frequency Compression (EFC) on the ғғтCᴏᴍᴘʀᴇss. It has a direct bearing on the ғғтCᴏᴍᴘʀᴇss's compression rate. Fig. 12 exhibits two key trends. Lower Q values compromise ARI and NMI due to aggressive data compression, causing the loss of essential information. On the other hand, elevating Q leads to performance plateaus or minor reductions, likely because of the introduction of noise.

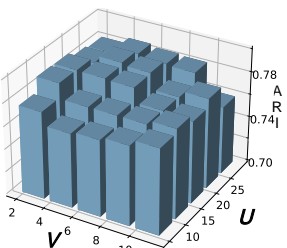

**Fig. 11: Impact of** U **and** V.

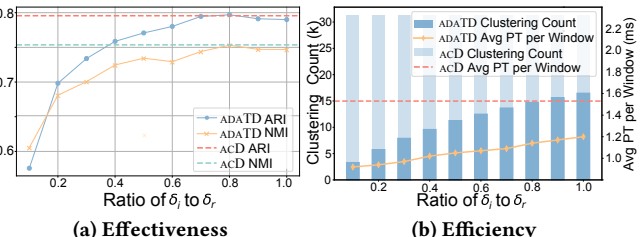

**Fig. 12: Impact of** Q.

*7.5.3 Impact of Intermediate Embedding Size* C *in* ᴅᴅEM. As illustrated in Fig. 13, enlarging the latent space dimensionality C generally boosts both ARI and NMI, peaking at C = 80. Further increases in C result in diminishing returns and even minor performance setbacks. Thus, by default, we set C to 80 for all the experiments.

*7.5.4 Impact of Final Embedding Size* D *in* ᴅᴅEM. As demonstrated in Fig. 14, increasing the embedding size D typically enhances ARI and NMI metrics, with a peak performance observed at D = 4. Beyond this value, the benefits of enlarging D decrease, and there

may even be slight performance deterioration. By default, we set D = 4 in E2Usᴅ.

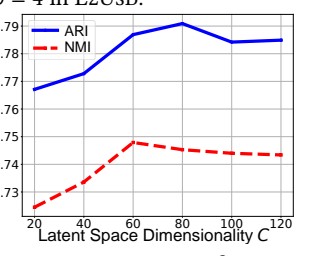
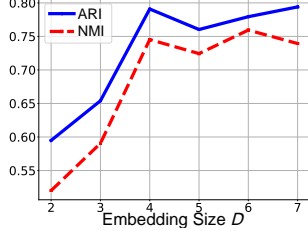

**Fig. 13: Impact of** C.        **Fig. 14: Impact of** D.

*7.5.5 Impact of* $\delta_i$ *and* $\delta_r$ *in* ᴀᴅᴀTD. In E2Usᴅ, the adaptability of ᴀᴅᴀTD stems from its ability to adjust the threshold $\tau$ based on the model's response, which is directly influenced by the $\frac{\delta_i}{\delta_r}$ ratio. For our evaluation, we set $\delta_r = 0.1$ and adjust the $\frac{\delta_i}{\delta_r}$ ratio within a range of 0.1 to 1.

As shown in Fig. 15, the effectiveness of ᴀᴅᴀTD improves incrementally with an increase in the $\frac{\delta_i}{\delta_r}$ ratio. Remarkably, it nears parity with the conventional "Always Clustering Detection" (ᴀcD) approach when $\frac{\delta_i}{\delta_r} = 0.8$. This is achieved while executing notably fewer clustering operations and maintaining a reduced average processing time per window.

(a) Effectiveness        (b) Efficiency

**Fig. 15: Impact of the ratio** $\delta_i$ **to** $\delta_r$.

