# OpenReview forum: "E2USD: Efficient-yet-effective Unsupervised State Detection for Multivariate Time Series"
_ACM.org/TheWebConf/2024/Conference — TheWebConf24 Oral_

### Official Review · Reviewer_4nTr · 2023-11-21

**Novelty:** 4
**Technical Quality:** 5

**Review:**

The paper aims to improve unsupervised state detection performance for multivariate time series (MTS) data. Existing methods suffer from high computational overhead, false negative sampling, and lack of optimization for online streaming scenarios. This paper proposes a method called E2USD to attempt to address these problems. E2USD leverages fftCompress and ddEM to reduce the computational cost, fnccLearning to improve the negative sampling, and adaTD to optimize the whole proposed solution for online streaming scenarios. The experimental analysis compares E2USD with several existing methods on six datasets, Synthetic, MoCap, ActRecTut, PAMAP2, UscHad, and UcrSeg, which shows the effectiveness of the proposed method.

**Pros**
1. This paper is well-organized with a clear structure and detailed explanation for core components, such as Section 2~3.
2. The component study and parameter sensitivity study are helpful to illustrate the effects of each component and parameter.
3. Intuitive figures, such as Figures 1~3, are provided to illustrate the key concepts.

**Cons**
1. The proposed method aims to improve the efficiency of unsupervised state detection for MTS. However, the experimental study lacks an in-depth analysis and explanation of why the proposed method can improve runtime efficiency.
2. The proposed method leverages contrastive learning, which lacks discussions in the related work.
3. Some important details are missing from this paper, such as how to set up the static threshold detection.

**Questions:**

Please check the cons listed above.

**Reviewer Confidence:**

3: The reviewer is confident but not certain that the evaluation is correct

**Scope:**

3: The work is somewhat relevant to the Web and to the track, and is of narrow interest to a sub-community

---

### Official Review · Reviewer_jrT4 · 2023-11-22

**Novelty:** 4
**Technical Quality:** 5

**Review:**

## Review

The paper presents an innovative unsupervised approach to state detection designed for environments with constrained computational resources. This method not only broadens the application spectrum but also addresses certain limitations inherent to current state-of-the-art (SOTA) techniques, particularly in practical deployments. The experimental outcomes support the method's efficacy, providing commendable results. The article's narrative structure is logically coherent, with each section seamlessly contributing to the overarching argument.

Pros:

- Effectively addresses false positives in contrastive learning with pairwise sample similarity.

- Reduced model usage with minimal accuracy loss via dynamic thresholding.

- Achieve performance precision nearing state-of-the-art methods.

Cons:

- The FFTCompress module functions like a low-pass filter in most scenarios. The necessity of calculating spectral intervals based on continuous spectral energy density seems questionable. This method's utility might be clarified by a more nuanced justification.

- The paper lacks detailed data presentation during practical validation on an MCU. Providing comprehensive experimental details would bolster the credibility of the results and allow for a clearer understanding of the method's performance in real-world settings.

**Questions:**

1. In real-world scenarios, what would be an efficient method to determine appropriate FFT bandwidth for the data?

2. Section 4.5 mentions an average sample latency of 44.95ms on the MCU. Could you specify what proportion of this delay is attributed to triggering cluster detection, and what are the actual latencies for both processes individually?

3. Typically in FFT analyses, the zero index contributes to a higher energy component, as shown in Figure 2(1) and Section 7.3. Given the method of selecting frequency bands described in the paper, wouldn’t a low-pass filter suffice in most cases instead of the FFTCompress module?

4. In Section 3.2, a lambda coefficient is introduced (below Equation 12) for selecting the least similar sample pairs. Does this coefficient significantly affect algorithm performance?

**Reviewer Confidence:**

3: The reviewer is confident but not certain that the evaluation is correct

**Scope:**

4: The work is relevant to the Web and to the track, and is of broad interest to the community

---

### Official Review · Reviewer_v6kn · 2023-11-24

**Novelty:** 4
**Technical Quality:** 5

**Review:**

The paper addresses the task of unsupervised identification of states in multivariate time series (MTS) within the scope of cyber-physical system sensors.

Pros:

- Novel Approach: The idea to combine the Fast Fourier Transform-based Time Series Compressor (fftCompress) and a Decomposed Dual-view Embedding Module (ddEM) is intriguing.

- Addressing Challenges: The paper addresses the challenges faced by existing state-detection methods, including computational overhead, false negatives, and a lack of emphasis on online streaming scenarios. The proposed Adaptive Threshold Detection (adaTD) for reducing computational overhead in streaming settings is a noteworthy contribution.

- Real-world Applications: The inclusion of comprehensive experiments with multiple baselines/datasets, along with testing on real devices and hardware, adds practical relevance to the proposed method. This application-driven approach enhances the paper's significance.

- Code Availability: The authors provide a link to the code, promoting reproducibility in the research community.

Cons:

- Negative Pair Mining: While the paper effectively tackles the challenges associated with negative pair mining, it could benefit from a discussion on alternative approaches that do without negative pairs, such as BYOL. This addition would contribute to the completeness of the paper's exploration of methodologies.

- Conceptual Differentiation: The paper references TCN but could provide a more detailed discussion on how the pair selection in E2Usd is conceptually different from methods like TCN. This clarification would help readers understand the unique contributions of E2Usd in relation to existing approaches.

- Ablation Analysis: Given the multiple proposed modules, an ablation analysis by removing components, such as FFTCOMPRESS, would be beneficial. This analysis could shed light on the individual importance of each module and provide insights into the system's robustness.

**Questions:**

In summary, the paper demonstrates a novel solution to the problem of unsupervised MTS state detection. It effectively addresses challenges and provides practical applications. To enhance the paper, I recommend further discussions on negative pair mining alternatives, conceptual differentiation from TCN, and an ablation analysis to assess the importance of individual components.

**Reviewer Confidence:**

3: The reviewer is confident but not certain that the evaluation is correct

**Scope:**

3: The work is somewhat relevant to the Web and to the track, and is of narrow interest to a sub-community

---

### Official Review · Reviewer_L7P1 · 2023-11-28

**Novelty:** 6
**Technical Quality:** 6

**Review:**

The paper addresses the challenge of unsupervised identification of states in multivariate time series (MTS) emitted by cyber-physical system sensors. It claims that their model is suitable for resource-constrained devices and streaming scenarios.
The proposed solution, E2Usd, utilizes a Fast Fourier Transform-based Time Series Compressor (fftCompress), a Decomposed Dual-view (Trend and Seasonality features) Embedding Module (ddEM), and a False Negative Cancellation Contrastive Learning method (fnccLearning). To reduce computational overhead in streaming scenarios, Adaptive Threshold Detection (adaTD) is introduced.
They have investigated the effectiveness of their method across 6 datasets compared to 6 other baselines.

The paper is very well written and easy to follow. The technical material and the rationale behind each decision are explained clearly.
The integration of fftCompress and ddEM for compact MTS embedding, along with the novel fnccLearning method, showcases originality in addressing the well-known challenges in unsupervised state detection. The adaptive threshold mechanism for streaming data further adds to the originality of the work.

**Questions:**

1. The Related work section and the baseline comparison do not cover the most related SOTA in the field of multivariate time series contrastive learning and False Negative Cancellation techniques.
    - CPC on dataset related to this work:

          -Harish Haresamudram, Irfan Essa, and Thomas Plötz. 2021. Contrastive predictive coding for human activity recognition. Proceedings of the ACM on Interactive, Mobile, Wearable and Ubiquitous Technologies5, 2(2021),1–26.

          - Shohreh Deldari, Daniel V. Smith, Hao Xue, and Flora D. Salim. 2021.Time Series Change Point Detection with Self-Supervised Contrastive Predictive Coding.In Proceedings of The Web Conference 2021 (WWW’21).
    - Contrastive learning on multivariate TS:

          - Shohreh Deldari, Hao Xue, Aaqib Saeed, Daniel V. Smith, and Flora D. Salim. 2022. COCOA: Cross Modality Contrastive Learning for Sensor Data. Proc. ACM Interact. Mob. Wearable Ubiquitous Technol.
     - False Negatives:

          - Huynh, T., Kornblith, S., Walter, M.R., Maire, M. and Khademi, M., 2022. Boosting contrastive self-supervised learning with false negative cancellation. In Proceedings of the IEEE/CVF winter conference on applications of computer vision (pp. 2785-2795).
          - Robinson, J., Chuang, C.Y., Sra, S. and Jegelka, S., 2020. Contrastive learning with hard negative samples. arXiv preprint arXiv:2010.04592.
         -  Jain, Y., Tang, C.I., Min, C., Kawsar, F. and Mathur, A., 2022. Collossl: Collaborative self-supervised learning for human activity recognition. Proceedings of the ACM on Interactive, Mobile, Wearable and Ubiquitous Technologies, 6(1), pp.1-28. >>> In this work, proposed for multi-device TS data, the authors proposed a novel technique based on Maximum Mean Discrepancy (MMD) to evaluate negative pairs.
          - etc.

2. The authors have considered the impact of false negatives and proposed an adaptive method to discard false negatives or reduce them. I was wondering what can the impact of False positives be. For example, in the streaming scenario, there might be a change in the state of the data. However, according to the random selection of positive pairs (i.e., consecutive frames), two positive pairs may not necessarily be related to the same distribution/state.

3. That would be great if the authors could do more experiments on evaluating their false negative cancellation method as one of their claimed contributions.


minor changes and typos:
 - the format of the text in the first and second paragraphs of Section 2.1 should not be italic.

**Reviewer Confidence:**

4: The reviewer is certain that the evaluation is correct and very familiar with the relevant literature

**Scope:**

3: The work is somewhat relevant to the Web and to the track, and is of narrow interest to a sub-community

---

### Decision · Program_Chairs · 2024-01-22

**Decision:**

Accept (Oral)

**Comment:**

The paper tackles the problem of detecting state from multivariate time series (MTS) data emitted by sensors. Reviewers appreciate that the paper was well written with all choices justified and motivated. There is also a commitment to open science including open source code repository being made available. The authors also engaged well with the reviewers' comments, including implementing a new baseline that applies BYOL to MTS data, which was pointed out by one of the reviewers. In any subsequent revision, the paper should include all the promised clarifications and changes that emerged during the rebuttal period, as it will improve the presentation significantly.